

# Wildfire switches the typical understanding of boreal peatland methane emissions

Scott J. Davidson[1], Christine Van Beest[1], Richard Petrone[1], Maria Strack[1]

[1] Department of Geography and Environmental Management, University of Waterloo, Waterloo, Ontario, Canada, N2L 3G1

*Correspondence to*: Scott J. Davidson (s7davidson@uwaterloo.ca)

**Abstract.** Boreal peatlands represent a globally important store of carbon, and disturbances such as wildfire can have a negative feedback to the climate. Understanding how carbon exchange and greenhouse gas (GHG) dynamics are impacted after a wildfire is important, especially as boreal peatlands may be vulnerable to changes in wildfire regime under a rapidly
changing climate. Yet, given this vulnerability, there is very little in the literature on the impact such fires have on methane ($CH_4$) emissions. This study investigated the effect of wildfire on $CH_4$ emissions at a boreal fen near Fort McMurray, AB, Canada, that was partially burned by the Horse River Wildfire in 2016. We measured $CH_4$ emissions and environmental variables (2017-2018) and $CH_4$ production potential (2018) in two different microform types (hummocks and hollows) across a burn severity gradient (unburned (UB), moderately burned (MB) and severely burned (SB)). Results indicated a switch in
the typical understanding of boreal peatland $CH_4$ emissions. For example, emissions were much lower in the MB and SB hollows in both years compared to UB hollows. Interestingly, across the burned sites, hummocks had higher fluxes in 2017 than hollows at the MB and SB sites. We found typically higher emissions at the UB site where the water table was close to the surface. However, at the burned sites, no relationship was found between $CH_4$ emissions and water table, even under similar hydrological conditions. This further strengthens the argument on the overriding influence of fire. There was also significantly
higher $CH_4$ production potential from the UB site than the burned sites. The reduction in $CH_4$ emissions and production in the hollows at burned sites highlights the sensitivity of hollows to fire, removing labile organic material for potential methanogenesis. The previously demonstrated resistance of hummocks to fire also results in limited impact to $CH_4$ emissions and likely faster recovery to pre-fire rates. Given the potential initial net cooling effect resulting from a reduction in $CH_4$ emissions, it is important that the radiative effect of all GHG following wildfire across peatlands is taken into account.

**1 Introduction**

Northern peatlands are an important component of the global carbon (C) cycle, acting as long-term sinks of atmospheric carbon dioxide ($CO_2$). They are also large sources of methane ($CH_4$) (Bridgham et al., 2013), with northern peatlands contributing approximately 40 – 155 Tg to global $CH_4$ emissions (Neef et al., 2010; Turetsky et al., 2014). $CH_4$ dynamics in peatlands results from a combination of various biogeochemical processes (Lai, 2009). Controls on $CH_4$ production, oxidation and
emissions include microtopography (Cresto Aleina et al., 2016), water table depth (Bubier et al., 1995; Granberg et al., 1997), soil temperature (Granberg et al., 1997; Saarnio et al., 1998), substrate quality and availability (Granberg et al., 1997; Segers, 1998; Joabsson et al., 1999), and vegetation cover (Ström et al., 2005; Strack et al., 2017).



However, disturbances such as wildfire can have a significant impact on the magnitude of C fluxes across peatlands (fire can release between 10-85 kg C m$^{-2}$ through combustion and smouldering; Turetsky et al., 2011), potentially causing a negative feedback to the climate (Randerson et al., 2006). Western boreal Canada is undergoing increasing pressure from wildfire, with fire extent and frequency expected to double by the end of this century (Benscoter et al., 2005; Flannigan et al., 2008).

Understanding how ecosystem C cycling and greenhouse gas (GHG) dynamics are impacted after a wildfire is important, especially as boreal peatlands may be vulnerable to changes in wildfire regime under a rapidly changing climate (Flannigan et al., 2008). Fire can remove surface vegetation, increasing net radiation at the ground surface (Brown et al., 2015), and can 're-set' vegetation communities back to the primary succession stage (Johnstone, 2006; Benscoter & Vitt, 2008). Fire can alter soil organic matter quality in the soil column (Neff et al., 2005; Olefeldt et al., 2013a) and reduce belowground C stores in

peatlands (Wilkinson et al., 2018). Overall, wildfire can lead to a decrease in C accumulation rate through combustion loss, reduction in vegetation productivity and increased organic matter decomposition post-fire (Ingram et al., 2019; Robinson & Moore, 2000; Wieder et al., 2009).

Microtopography across peatlands can be impacted through fire, by increasing the prominence of hollows on the landscape through altering elevation (Benscoter et al., 2015), and often hollows will have a higher severity of burn compared to other

areas across the landscapes (Mayner et al., 2018; Benscoter et al., 2005). Conversely,  hummocks are generally resistant to fire, namely due to moisture retention differences between the different moss species present at both microform types, as *Sphagnum* spp. is much more resilient to fire than feather moss (Kettridge et al., 2015).  Reduction in vegetation cover and soil organic matter can also lead to drier conditions across peatlands (Tarnocai, 2009; Thompson & Waddington, 2013; Kettridge et al., 2015), with the drop in water table level causing an increase in the aerobic zone (Waddington et al., 2015).

This could lead to a reduction in $CH_4$ emissions or even uptake of $CH_4$ via oxidation (Strack et al., 2004; Turetsky et al., 2008; Moore et al., 2011). Conversely, high water tables can occur post-fire (Kettridge et al., 2015), although often associated with low surface moisture contents due to hydrophobicity of the peat (Doerr et al., 2000).

Despite the increasing pressures from wildfires across northern peatlands, a knowledge gap still persists on $CH_4$ emissions after wildfire, especially in boreal regions. In a study on the impact of wildfire on methanotrophic communities from an

ombrotrophic peat bog, Danilova et al. (2015) found a reduction in the activity of the methanotrophs in burned sites 7 years post-fire. This reduction following wildfire could therefore  lead to a potential increase in $CH_4$ emissions from bog systems. Grau-andrés et al. (2019) also showed an increase in $CH_4$ emissions at an ombrotrophic bog in the UK one year after a prescribed fire, most likely due to increased graminoid coverage. Conversely, studies at other bog sites in the UK report a decrease in emissions after fire (Ward et al., 2007; Davies et al., 2013). In non-peatland ecosystems across boreal regions,

wildfire has been shown to cause an increase in $CH_4$ uptake (Burke et al., 1997; Song et al., 2017, 2018).  However, in permafrost zones, wildfires can often typically lead to substantial permafrost thaw and increasing moisture levels across the landscape (Gibson et al., 2018), potentially leading to an increase in $CH_4$ emissions (Kim & Tanaka, 2003; Turetsky et al., 2008; Olefeldt et al., 2013b; Helbig et al., 2017). However, Köster et al. (2017; 2018) found an increase in $CH_4$ uptake across




a continuous permafrost site in both Canada and Russia after fire. To date, we cannot find any reference on the impact of fire on $CH_4$ emissions across fens, despite being the dominant peatland type in western boreal Canada (Vitt et al., 2000).

Therefore, the objectives of this study are to: i) determine the impact of wildfire on fen $CH_4$ emissions across a burn severity gradient; ii) evaluate the controls on $CH_4$ emissions within each site; and iii) examine $CH_4$ production potential across a burn

severity gradient. We hypothesize that $CH_4$ emissions and production will be lower at burned sites due to lowering of the water table, changes in substrate availability, and reduction in vegetation cover.

## 2 Methods

### 2.1 Study site and collar locations

The study was undertaken in a fen (hereafter referred to as Poplar Fen), near Fort McMurray, Alberta, Canada, which was

partially burned by the Horse River Wildfire in 2016. Poplar Fen is a treed, moderate-rich fen located approximately 20 km north of Fort McMurray (56°56.330 N, 111°32.934 W). The mean annual temperature (1981-2010) is 1 °C, and mean annual precipitation is approximately 420 mm (Environment Canada, 2017). This treed fen is dominated by *Larix laricina, Picea mariana, Betula pumila, Equisetum fluviatile, Smilcina trifolia, Carex* spp. and *Sphagnum fuscum* and brown mosses, largely *Tomenthypnum nitens*. Average peat depth ranges between 1 and 1.5 m. This site was split into three sections along a peat burn

severity gradient as assessed by depth of burn (DOB). The unburned (UB) site was situated within the fen interior and was not affected by the wildfire. The moderately burned (MB) site had a DOB of approximately 9 – 11 cm and the severely burned (SB) site had a DOB of approximately 14.5 – 17 cm. Both burned sites were situated on the peatland margin, closer to the adjacent upland at a slightly higher elevation. The DOB was determined following the protocols used by Lukenbach et al., (2015a), van Beest et al. (In Review).  At each site, PVC collars (height 15 cm x diameter 20 cm) were placed in both hollows

and hummocks. A total of four replicate collars were placed at each microform location at each site to a depth of approximately 15 cm in Spring 2017, totaling eight collars at the UB site, eight collars at the MB site and eight collars at the SB site. The height of the collar was measured from the soil surface in order to have the correct chamber headspace volume for $CH_4$ emission calculations.

### 2.2 Environmental Conditions

Water table (WT) depth (relative to the ground surface) was measured adjacent to each pair of collars at all three sites. A PVC pipe (4 cm (diameter) x 100 cm (length)) fully slotted along the full length and covered in mesh was used. A soil temperature (ST) profile was collected at each collar during each $CH_4$ measurement at -30, -25, -20, -15, -10, -5, -2 cm from ground surface. Percentage cover of plant functional type (bryophyte, graminoid, dwarf shrub), as well as bare ground, burned material and

standing water was estimated from photographs taken at peak growing season during 2018 to produce percentage cover estimates of the flux collars.



### 2.3 Measurements of field CH₄ emissions

Methane fluxes were measured using the closed chamber method, eight times between 7th May and 16th August 2017 and 14 times between 11th May and 16th August 2018. A cylindrical opaque chamber (20 cm x 50 cm) was placed on the collar, with water poured around the collar edge to create a seal. A battery powered fan was used to mix the chamber headspace. A

thermocouple located within the chamber, attached to a thermometer was used to measure temperature during sampling. A 20 mL syringe was used to collect gas samples at intervals of 7, 15, 25 and 35 minutes following chamber closure and injected into Exetainers (Labco, UK). A gas chromatograph (GC; Shimadzu GC2014, Mandel Scientific, Canada) with a flame ionization detector (250 °C), helium gas carrier and standards of 5 and 50 ppm was used to determine $CH_4$ concentration of the gas samples collected during the field seasons. The emissions were determined from the linear change in concentration

over time, which includes corrections for temperature and volume of the chamber. Any small negative or positive values in which the change in concentration did not exceed 10% (precision of concentration analysis) were assigned a zero-emission value. Large negative emissions ($< -5$ mg $CH_4$ $m^2$ $d^{-1}$) were removed from the analysis as these emissions were likely caused by disturbance during chamber placement. These procedures resulted in a total loss of 6 % of the data across both years.

### 2.4 Potential CH₄ production

Peat samples were collected adjacent to each collar (2-3 m away to avoid disturbing the collar) on the 13th August 2018 and immediately shipped to the laboratory and frozen until analyzed. Peat samples were obtained using a tin can to a depth of 20 cm from the ground surface (two samples were collected per sampling location; 0 – 10 cm and 10 – 20 cm).

Potential $CH_4$ production was determined under anaerobic conditions following a similar methodology to Strack et al., (2004).

Peat slurries of approximately 20 g of wet peat were made in 250 mL incubation jars. Distilled water was added to the sample to saturate, without allowing for standing water. Samples were flushed with $N_2$ for 15 minutes and then sealed. Slurries were incubated at room temperature (approximately 20°C) and sampled at 0, 24, 48 hr and then twice weekly between 16th October 2018 and 2nd November 2018 (total incubation length of 17 days). Samples were agitated by hand before sampling commenced to mix the gases within the peat pore spaces and the jar headspace.

Samples (10 mL) were extracted from the jars and injected into a Fast Methane Analyzer (FMA; Los Gatos, USA). A 10 mL sample of $N_2$ was replaced in each jar after the sample was extracted to maintain headspace pressure. Potential methane production was determined from the linear increase in $CH_4$ concentration within the jars over the incubation period after correcting for dilution by $N_2$ (Strack et al., 2017). Gravimetric soil moisture (GWC) was determined by weighing subsamples of peat not used in the incubation, drying the samples at 60°C for 2-3 days and reweighing. Organic matter content was

determined by Loss on Ignition (LOI), burning samples at 550°C for 4 hours.

### 2.5 Data analysis

All statistical analysis was undertaken in R (R Core Team 2013) using the package *nlme* (Pinheiro et al., 2018), and all output and models were inspected for normality and homogeneity of residuals (Zuur et al., 2009). Data were log transformed if




required, with a value of ten being added prior to transformation to account for zeros and negative values in the dataset. Seasonal mean values of $CH_4$ flux and associated environmental variables at each collar were used in all model levels (Treat et al., 2007; Turetsky et al., 2014) with statistical significance considered at the $\alpha = 0.05$.

In order to evaluate the effect of burn severity on $CH_4$ flux, a linear mixed effects model (LMM) was used with burn severity,
microform, the two-way interactions between these, and year as fixed factors, with collar ID as a random effect to take into account repeated measures (Pinheiro et al. 2018). Another LMM was used to evaluate the environmental controls on $CH_4$ flux, with the effect of burn severity, WT, ST at 30 cm depth and the two-way interactions between each included as fixed effects. If significant factors were found, Tukey pairwise comparisons were completed using the lsmeans package (Lenth, 2016). Any insignificant factors were removed from the model until the final model was found. An individual insignificant factor was kept
in the model if its interaction with another factor was significant. The amount of variance explained by the model ($R^2_{GLMM}$) was calculated using the method described by Nakagawa & Schielzith (2013).

A two-way Analysis of Variance (ANOVA) was used to test for differences between $CH_4$ production rate, burn severity and microform. No significant difference between sampling depth was found, thus depths were combined in further analyses. Again, if significant differences were found, Tukey pairwise comparisons were completed.

**3 Results**

**3.1 Environmental and vegetation variables**

Water table depth was linked to microtopographic position, with hollows having highest WT position across all sites. The deepest WT depth (± standard deviation) during both 2017 and 2018 was found at the SB hummocks, approximately -45 ± 5.6 and -44 ± 6.0 cm below the surface respectively (Table 1). The shallowest WT was found at the unburned hollows, being -8.6
± 4.1 and -6.9 ± 1.8 cm below the surface in 2017 and 2018, respectively (Table 1). Average ST at 30 cm depth (taken as spot measurement during each gas flux measurement) was similar across the burn severity gradient at approximately 10 – 12 °C (Table 1).

The vegetation survey of the collars undertaken in 2018 indicated that bryophytes dominated across all collars at all sites, regardless of microtopographic position. UB hollows were dominated by the moss *T. nitens*, while the hummocks were
dominated by *S. fuscum*. The SB hummocks and hollows both had the highest percentage of bare ground at ~ 49% and ~ 55% cover respectively, indicating vegetation (mostly *T. nitens*) was completely removed during the fire. It was noted that *Polytrichum strictum* moss was beginning to colonise these bare areas. The MB hummocks had the highest percentage of burned material (~ 40 % cover; predominantly singed *S. fuscum*) (Figure 1).

**3.2 $CH_4$ emissions**

The average $CH_4$ flux (± standard deviation) at unburned hollows was 126.5 ± 80.5 and 56.3 ± 18.9 mg $CH_4$ $m^{-2}$ $d^{-1}$ in 2017 and 2018, respectively (Figure 2). $CH_4$ emissions were much lower in the MB and SB hollows in both years, with the average flux being -0.38 ± 1.6 and -0.46 ± 0.9 mg $CH_4$ $m^{-2}$ $d^{-1}$ in 2017 and 0.21 ± 1.7 and 0.62 ± 2.5 mg $CH_4$ $m^{-2}$ $d^{-1}$ in 2018 (Figure



2), respectively. Interestingly, across the burned sites, hummocks had higher fluxes in 2017 than hollows with the average flux being $1.10 \pm 2.04$ mg $CH_4$ m$^{-2}$ d$^{-1}$ at the MB site and $4.53 \pm 9.3$ mg $CH_4$ m$^{-2}$ d$^{-1}$ at the SB site (Figure 2). During 2018, fluxes were lower, with hummocks being a slight sink of $CH_4$ with average fluxes of $-0.18 \pm 2.06$ mg $CH_4$ m$^{-2}$ d$^{-1}$ at the MB site and $0.43 \pm 1.6$ mg $CH_4$ m$^{-2}$ d$^{-1}$ at the SB site (Figure 2). To determine whether emissions measured from the UB site were

5 representative of emissions from Poplar Fen as a whole, we compared our fluxes to a previous study of $CH_4$ emissions collected between 2011 and 2014 at the fen. No significant difference in fluxes was found [t (22.5) = 1.5, p = 0.154] (Figure S1). Results of the LMM illustrate that there was a significant effect of burn on $CH_4$ flux (Table 2), but no significant effect of microform or year (Table 2). The second LMM considering environmental controls explained 63% of the variance in $CH_4$ emissions and found a significant interaction between burn severity and WT depth (Table 2; Figure 3), but no significant

relationship between $CH_4$ flux with WT depth, ST (Figure S2) or burn severity alone.

### 3.3 Potential $CH_4$ production

Measured potential $CH_4$ production was highest in the unburned hollows ranging from between 0.006 and 0.13 µg g$^{-1}$ peat hr$^{-1}$ (Figure 4); however, there was no significant effect of burn severity [ANOVA, F = 2.959, p = 0.065]. $CH_4$ production was

15 much lower across the burned sites, ranging between 0.0001 and 0.004 µg g$^{-1}$ peat hr$^{-1}$. The MB hummocks followed a similar pattern to the field measurements of $CH_4$ flux, having higher potential $CH_4$ production than the hollows. No significant difference in organic matter content or gravimetric water content was found between sites or microform types (Table S1).

### 4 Discussion

Fire had a strong effect on $CH_4$ emissions in this study, causing a large decrease in $CH_4$ flux in the MB and SB hollows in comparison to the UB hollows. Conversely, this study also highlights the resistance of hummocks to fire (Wieder et al., 2009; Benscoter et al., 2015), with hummocks across the burned sites maintaining higher $CH_4$ emissions after the fire compared to hollows. Methane production in the laboratory followed a similar trend to the field study, with highest production in the UB hollows and virtually no production in the burned hollows, again highlighting this reversal of typical peatland $CH_4$ emissions.

These results contrast with other studies looking at $CH_4$ emissions post-fire at peatland sites, with both Danilova et al. (2015) and Grau-andrés et al. (2019) indicating that fire across an ombrotrophic bog could decrease $CH_4$ oxidation due to removal of the methanotrophic community and potentially increase $CH_4$ emissions due to increased graminoid cover. We do not specifically measure $CH_4$ oxidation in this study.

We hypothesized that the lower $CH_4$ emissions at the burned sites could be due to the intensity of the burn, reducing substrate

availability (labile carbon) and minimizing the methanogenesis community, resulting in lower emissions. An increase in fire frequency has the potential to reduce organic matter quality and change vegetation communities in peatlands (Lukenbach et al., 2015b). Associated with a change in vegetation communities is the potential change in biogeochemical cycling and microbial processes (Ward et al., 2007). For example, the slight recovery in $CH_4$ emissions in 2018 (two years postfire) could be due to vegetation recovery (Ward et al., 2007), providing more available substrate through root exudates for $CH_4$ production





(Greenup et al., 2000; Robroek et al., 2015). An addition of graminoid cover in the SB hollows post-fire was also found, which could also lend itself to increasing $CH_4$ emissions in the future, as plant-mediated transport of $CH_4$ is well documented across peatland ecosystems (Bellisario et al., 2016).

Higher emissions at the UB site could result from overall shallower WT at this location compared to the MB and SB sites (Table 1), which were located at the fen margins. This topographic setting likely contributed to the limited effect of the wildfire at this location, but could also result in higher $CH_4$ emissions than would have occurred naturally at the burned sites prior to the fire. However, the comparison of our results to emissions measured between 2011 and 2014 at another location in Poplar Fen burned during the fire indicate there was no significant difference in $CH_4$ emissions.

Interestingly, we see no relationship with $CH_4$ emissions and WT depth at the burned sites. This switch in the typical understanding of the relationship between $CH_4$ emissions and WT further strengthens our argument on the overriding influence of fire. Even under suitable hydrological conditions, there is a lack of $CH_4$ production, as shown in the incubation study.

The higher $CH_4$ production found at the MB hummocks is likely due to the small methanogen community surviving the fire, due to the resistance of *S. fuscum* (Benscoter et al., 2011). After fire, there could be chemical changes, such as an increase in availability of terminal electron acceptors, that could contribute to the reduction in $CH_4$ production and emissions (Wilson et al., 2017). Therefore, there is a potential long-term impact on the biogeochemical processes of peatlands (Danilova et al., 2015) and in order to fully understand the overall impact of wildfire on $CH_4$ emissions, additional studies at other sites encompassing the full range of boreal peatland types would be key. This is especially true given the conflicting results in the literature regarding the overall impact of fire across a variety of peatland sites. Continuous monitoring of the recovery of the ecosystem over time could help evaluate whether $CH_4$ emissions return to similar levels as the undisturbed site.

## 5 Conclusion

This study investigated the impact of wildfire on $CH_4$ emissions at a treed, moderate-rich fen in northern Alberta. We believe this is the first study to investigate the impact of wildfire on $CH_4$ emissions at a non-permafrost boreal fen. The results showed a significant impact of fire on the magnitude of $CH_4$ flux, with a significant reduction in flux observed at the burned sites in comparison to the unburned (UB) site. No relationship was found with water table at the burned sites, contrasting the significant relationship at the UB site, further illustrating that methanogenesis was limited following fire. This was further supported by a lower rate of $CH_4$ production from peat collected at burned sites compared to UB, likely linked to reduced methanogen population and/or substrate availability due to the resistance of *Sphagnum* spp. hummocks reducing burn severity. With the expected increase in wildfire frequency across western boreal Canada, it is vital we fully understand the impact of fire on $CH_4$ dynamics. If fire is to reduce $CH_4$ emissions and production across these peatland ecosystems, there is a potential initial net cooling effect, therefore it is important to take into account the radiative effect of all GHG following wildfire.



## Acknowledgments

Funding for this project was provided by a Natural Sciences and Engineering Research Council of Canada (NSERC) Collaborative Research and Development (CRD) grant to MS and RP co-funded by Suncor Energy Inc., Imperial Oil Resources Limited, Teck and Shell Canada Energy and an NSERC Discovery Grant awarded to MS. The authors would like to

acknowledge Canada's Oil Sands Innovation Alliance (COSIA) for its support of this project. We thank J.M. Waddington and M. Helbig for helpful comments on an earlier version of the manuscript. Finally, we'd like to thank Matthew Coulas, Mariah Smith, Dryden Miller and Emily Prystupa for their help in the field.

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



**Table 1. Environmental characteristics (mean (± standard deviation)) for each microform type across burn severity gradient.**

| Year | Burn severity [a] | Microform | Water table depth (cm) | Soil temperature at 30 cm depth (°C) |
|---|---|---|---|---|
| | UB | Hollow | -8.6 (4.1) | 11.0 (1.1) |
| | | Hummock | -19.9 (5.5) | 11.5 (3.6) |
| 2017 | MB | Hollow | -20.9 (7.6) | 11.2 (3.4) |
| | | Hummock | -37.5 (9.2) | 10.6 (1.7) |
| | SB | Hollow | -24.0 (15.3) | 10.2 (2.0) |
| | | Hummock | -45.6 (5.6) | 9.9 (2.7) |
| | UB | Hollow | -6.9 (1.8) | 12.1 (1.9) |
| | | Hummock | -22.6 (6.6) | 12.3 (1.3) |
| 2018 | MB | Hollow | -9.0 (0.9) | 11.1 (1.1) |
| | | Hummock | -23.6 (0.2) | 12.8 (0.8) |
| | SB | Hollow | -14.2 (1.0) | 10.3 (0.3) |
| | | Hummock | -44.5 (6.0) | 11.4 (1.4) |

[a] UB is unburned, MB is moderately burned and SB is severely burned.



**Table 2. Statistical results of the linear mixed effects model [a]**

| Flux component | Effect | F | P | $R^2_{GLMM}$ [b] |
|---|---|---|---|---|
| | Burn severity | $F_{1,22} = 83.02$ | < 0.0001 | |
| | Microform | $F_{2,17} = 2.4$ | 0.14 | |
| | Year | $F_{1,22} = 1.2$ | 0.3 | 0.52 |
| | Burn severity × Microform | $F_{2,17} = 5.5$ | 0.014 | |
| | Intercept | $F_{1,22} = 2075.3$ | < 0.0001 | |
| $CH_4$ [c] | Burn severity | $F_{2,20} = 83.4$ | < 0.0001 | |
| | WT depth | $F_{1,17} = 0.5$ | 0.5 | |
| | ST at 30 cm depth | $F_{1,17} = 3.0$ | 0.1 | |
| | Burn severity × WT depth | $F_{2,17} = 3.3$ | 0.046 | 0.63 |
| | Burn severity × ST at 30 cm depth | $F_{2,17} = 3.4$ | 0.06 | |
| | Intercept | $F_{1,20} = 2085.4$ | < 0.0001 | |

[a] All models have a random factor of collar ID to take into account the repeated measures across both years

[b] We report the marginal $R^2_{GLMM}$ accounting for variance explained by fixed factors only

[c] The model was calculated using $\log_{10} CH_4$ values




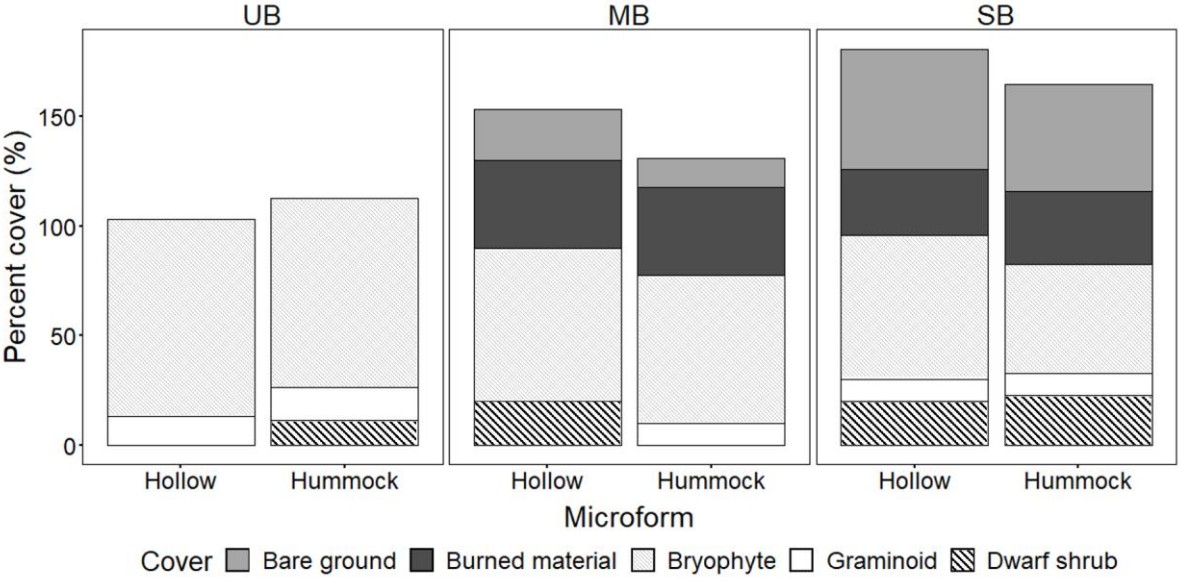

**Figure 1. Vegetation cover (%) for each dominant plant functional type and ground cover for the flux collars in 2018.**

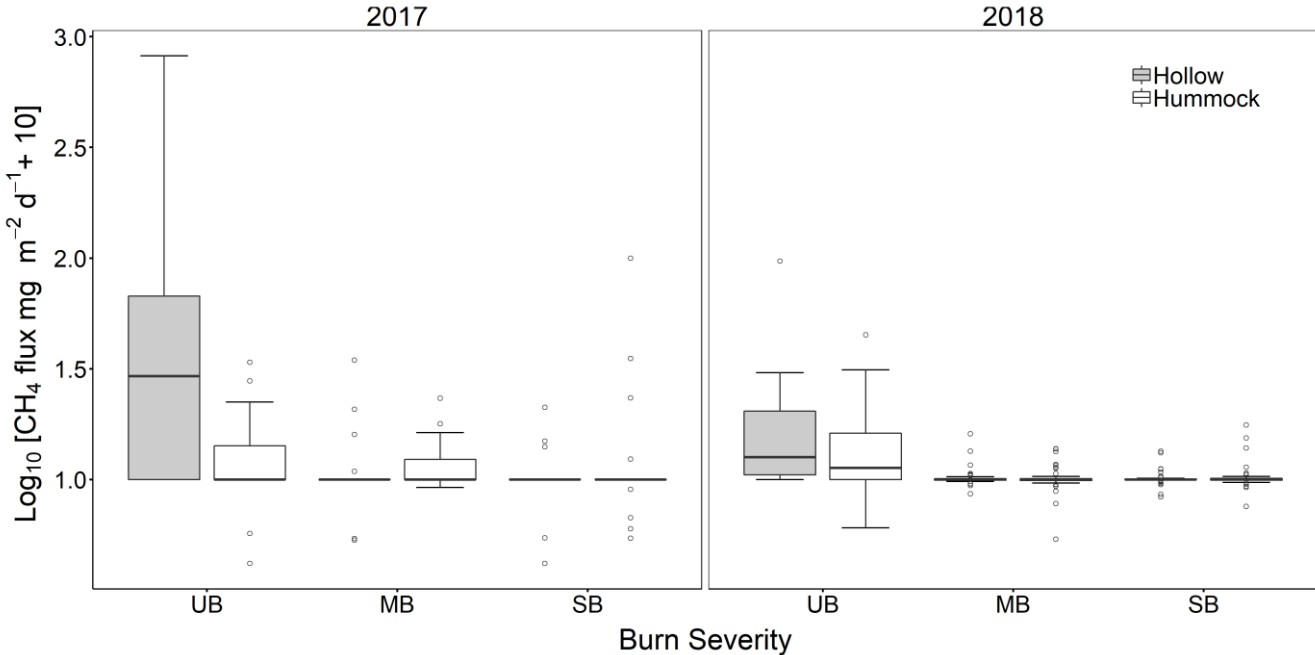

**Figure 2. Methane (CH$_4$) emissions at each microform type across the peat burn severity gradient for 2017 and 2018. UB is unburned,**
5  **MB is moderately burned and SB is severely burned. Note that CH$_4$ values were log transformed + 10 therefore a value of 1 represents the measured value zero.**





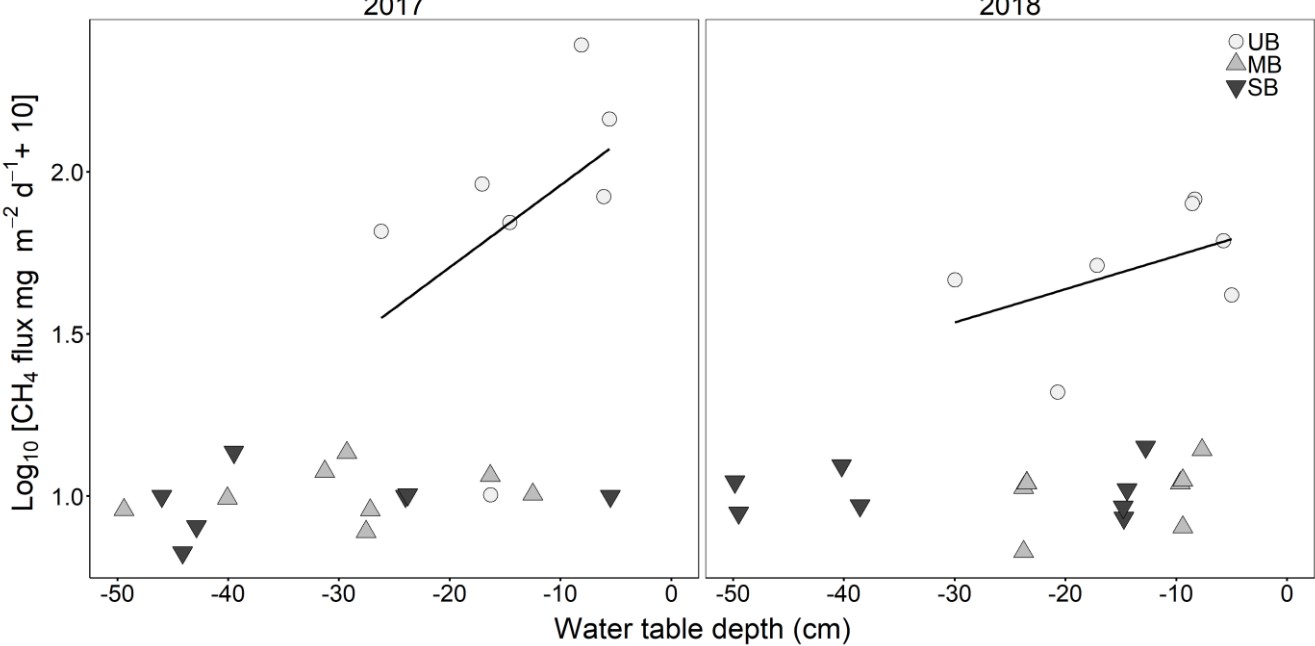

**Figure 3. Seasonal mean methane (CH₄) flux at each collar across the peat burn severity gradient plotted against seasonal mean water table (WT) depth. UB is unburned, MB is moderately burned and SB is severely burned. Note that CH₄ values were log transformed + 10 therefore a value of 1 represents the measured value zero.**



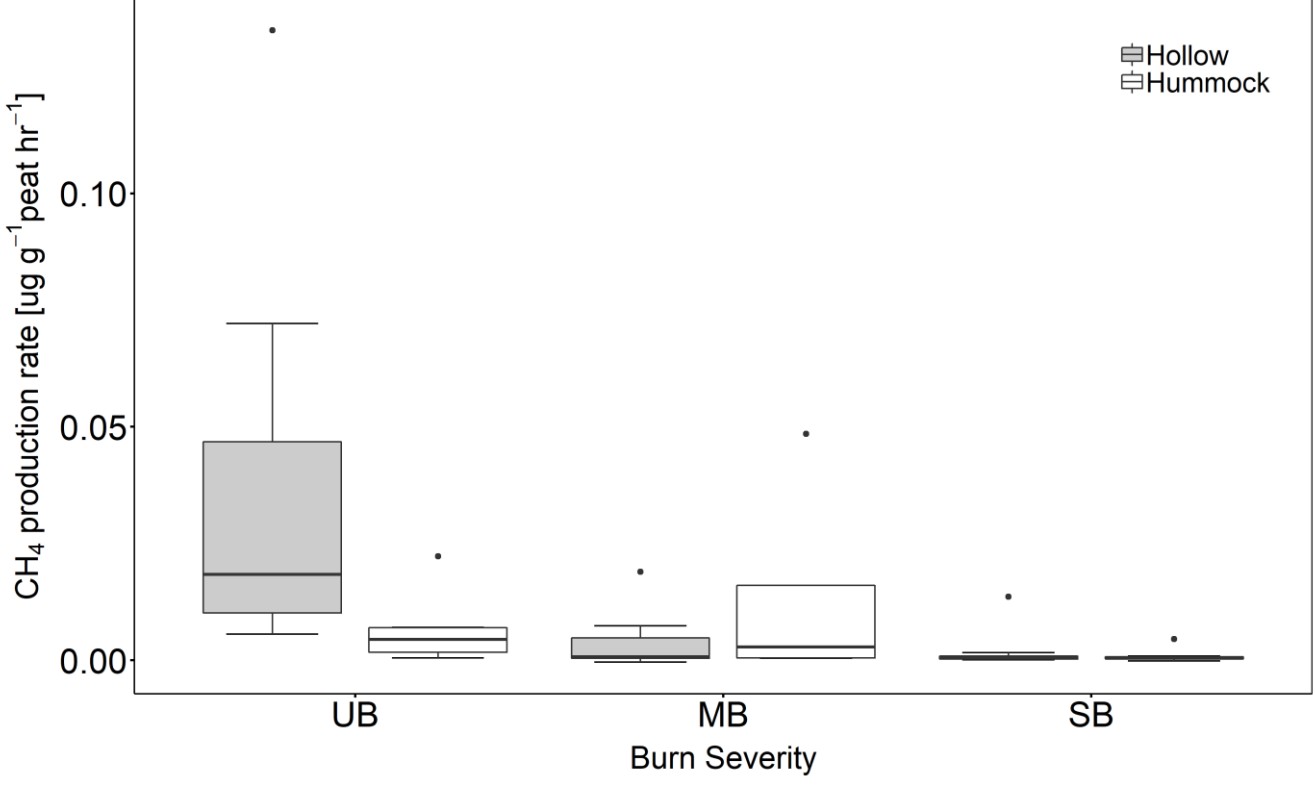

**Figure 4. Potential methane (CH₄) production across the peat burn severity gradient and microform type. Each microform represents four sample replicates.**