# Peer review of "Wildfire overrides hydrological controls on boreal peatland methane emissions"

_Biogeosciences, 2019_

## Referee Comment (RC1) · Anonymous Referee #1 · 20 May 2019

Authors of the manuscript have studied the topic of utmost importance in the field of environmental research and the topic of their research is within the scope of current journal. In the World with warming climate and accompanying increased fire activity in boreal forest and peatland areas, more attention should be payed to the topic. Methane emissions of boreal fen are so far poorly studied and current manuscript has significant results to fill this cap. The text of the manuscript is fluent and easy to follow. The scientific quality of the research is excellent: experiment planning, data collection and data analysis support that received results have scientific value. I find that the manuscript can be considered for publication after minor revision. My specific comments on the manuscript are following:

Specific comments.

[Figure]

Abstract: Page 1, line 15: Could you be more specific: were the emissions in the MB and SB significantly lower when compared to UB.

Introduction Page 2, line 2: A bracket is missing. Page 2, lines 5-13: Could you add also some information about the fire induced deposition of charcoal and ashes, and their effect to the soil pH and physical characteristics. Page 2, line 22: You mention the hydrophobicity of the peat as the reason for low surface moisture content. What about deposited ashes?

Methods Page 3, line 12-14: Please provide Latin names of plants and mosses with proper affiliations. This comment goes for all Latin names mentioned in the text. Page 3, line 15-19: Can you assure that the initial conditions of peatland margin and interior part were comparable by means of vegetation cover and water table conditions? Page 3, line 29-31: What about the vegetation succession? How much did the vegetation cover change between 2017 and 2018? Could this be the reason for changed CH4 fluxes? Page 4, line 5-7: Why did you start with the gas collections 7 minutes after chamber was placed? Page 5, line 7: You have used the soil temperatures measured at 30 cm depth, although measurements were done from 2-30 cm depth. Is there any scientific explanation for that?

Results: Page 6, line 4-6: Please move this sentence to the Material part. This is partly explaining my question about the conditions in peatlands interior and margin areas.

Discussion: Page 7, line 2: "lend itself to" sound like informal language to me, but as I am not a native speaker, I might be also mistaken. Page 7, line 19: I would also add the time factor to the sentence. I think "whether" the CH4 emissions will return to similar levels is rather sure. It is much more interesting how long it takes. . .

Figure 2: Graphics of current figure do not able to understand the microform type for most of burn severity classes.

Please also note the supplement to this comment:

https://www.biogeosciences-discuss.net/bg-2019-139/bg-2019-139-RC1-supplement.pdf

---

## Referee Comment (RC2) · Anonymous Referee #2 · 29 May 2019

Davidson and colleagues present a study in which variability in methane emissions have been quantified for different burn severity classes and microtopographic positions in a boreal fen in Canada. The research begins to answer important questions regarding interactions between fire disturbance and methane cycling in the context of boreal carbon cycle feedbacks to climate. The authors find that fire generally reduces methane emissions and, for at least several years following fire, eliminates relationships between water table depth and methane emissions. The paper is based on a relatively small but important data set that is appropriately analyzed. The paper will be suitable for publication after a few relatively minor revisions. I have several overarching comments followed by more specific ones.

This manuscript is rather short, which isn't a bad thing, however I do think there is room

to expand and add some additional details, especially in the discussion. For example, permafrost is invoked as a potentially important driver in the Introduction, but then is not mentioned in the discussion. The main point regarding the wildfire overriding hydrological controls on CH4 emissions comes through. But the secondary point that these effects are understudied and may vary with ecosystem type could be developed more.

I would encourage the authors to archive their data and code in open access repositories, ideally where they would be citable with a doi.

It is a little confusing to have CH4 flux numbers reported normally in the text of the results, and then log-transformed in the figures. It would be best to back-transform for the figures if possible.

P1 L1: Different disciplines may understand peatland methane emissions in different ways. Why not use "Wildfire overrides hydrological controls on boreal peatland methane emissions" or something similarly specific?

P1 L8: Are you referring specifically to a negative carbon cycle feedback to climate, or any negative feedback (e.g. surface energy partitioning)

P1 L19: This sentence seems a little odd since you have not yet mentioned any argument regarding the overriding influence of fire. Maybe reword, or set it up better.

P2 L13: It would be nice to have the briefest of descriptions of hummocks and hollows, I think I know what they are. Also what is the mechanism by which fire alters elevation, causing hummocks. Lastly, it would be interesting to know how prevalent hummocks vs. hollows are on the landscape – are there any papers out there with numbers you could site (e.g. hummocks make up XX% and hollows XX% of typical fen's in this region).

P2 L17: Can you say specifically which moss species is found at each microform type?

P2 L31: Is your site underlain by permafrost? I don't think it's reported in the methods.

That could be useful to know.

P3 L1: A single site in both Canada and Russia?

P3 L18: Please give a one or two sentence description of the DOB protocol. The reader should only be obligated to look at other refs if they want all of the juicy details.

P30 L30: Is it possible that these cover variables could have changed of the course of your study? Particularly moss colonization, but also water, which I imagine could change with the weather conditions. Also, could you briefly describe your percent cover approach; since some sites have more than 100% I assume you are looking at over story and ground cover?

P4 L12: Please include justification or reasoning for the -5 mg Ch4/m2/d threshold.

P6 L25: Do both of these studies indicate both of these things, or is each point from one of the studies?

P7 L1: What do you mean by an addition here?

P7 L4: Please expand the discussion of fire effects on water table depth. A more in-depth process level discussion would be nice here, perhaps also with some specifics on the variability within this fen that you allude to.

P7 L13: The resistance of S fuscum to what?

P7 L14: Chemical changes in the soil substrate? Can you be a little more specific here?

---

## Author Comment (AC1) · 4 Jun 2019

**We thank the reviewer for their constructive comments on our manuscript. Here we provide our initial responses to these comments and will provide a modified manuscript after the discussion is closed.**

Title: Wildfire switches the typical understanding of boreal peatland methane emissions.
Authors of the manuscript have studied the topic of utmost importance in the field of environmental research and the topic of their research is within the scope of current journal. In the World with warming climate and accompanying increased fire activity in boreal forest and peatland areas, more attention should be payed to the topic. Methane emissions of boreal fen are so far poorly studied and current manuscript has significant results to fill this cap. The text of the manuscript is fluent and easy to follow. The scientific quality of the research is excellent: experiment planning, data collection and data analysis support that received results have scientific value. I find that the manuscript can be considered for publication after minor revision.

**We thank the reviewer for their positive comments about our study.**

My specific comments on the manuscript are following:

Specific comments.
Abstract:
Page 1, line 15: Could you be more specific: were the emissions in the MB and SB significantly lower when compared to UB.

**We have changed the text; Pg 1, new line 15-16**

**"For example, emissions were significantly lower in the MB and SB hollows in both years compared to UB hollows"**

Introduction
Page 2, line 2: A bracket is missing.

**The open bracket is found on Pg 2, new line 1 : "(fire can release between 10-85 kg C m$^{-2}$ through combustion and smouldering; Turetsky et al., 2011),"**

Page 2, lines 5-13: Could you add also some information about the fire induced deposition of charcoal and ashes, and their effect to the soil pH and physical characteristics.

**We have included reference to the impact of ash deposition on soil pH and physical characteristics on Pg 2 lines 12-15. Furthermore, we have moved the paragraphs around for clarity.**

Page 2, line 22: You mention the hydrophobicity of the peat as the reason for low surface moisture content. What about deposited ashes?

**We have now included reference to the impact of increased ash deposition after fire on soil moisture on Pg 2, new lines 19-20.**

**"(Doerr et al., 2000). Low soil moisture rates can also occur under increased ash deposition after fire, with increased closure of soil pores by ash causing reduced capacity to hold water and increased runoff (Heydari et al., 2017)."**

Methods
Page 3, line 12-14: Please provide Latin names of plants and mosses with proper affiliations. This comment goes for all Latin names mentioned in the text.

**We have included the latin names and proper affiliations of the plant species on Pg 3, new lines 16-20;**

**"This treed fen is dominated by Larix laricina (Du Roi) K.Koch, Picea mariana (Mill,) Britton, Betula pumila (L.), Equisetum fluviatile (L.), Smilacina trifolia (L.) Sloboda, Carex spp. and Sphagnum fuscum (Schimp.) Klinggr and brown mosses, largely Tomenthypnum nitens (Hedwig) Loeske."**

Page 3, line 15-19: Can you assure that the initial conditions of peatland margin and interior part were comparable by means of vegetation cover and water table conditions?

**Although we understand that there are slight differences in environmental conditions between the peatland margins and centre, we do believe that we can successfully compare the two. Furthermore, we now include reference to a comparison study we undertook between methane emissions collected between 2011 and 2014 that was representative of what burned in 2016.**

**See new Pg 4 new lines 23-25: "In order to determine whether emissions measured from the UB site were representative of emissions from Poplar Fen as a whole, we also compared our fluxes to a previous study of CH4 emissions collected between 2011 and 2014 at the fen.".**

Page 3, line 29-31: What about the vegetation succession? How much did the vegetation cover change between 2017 and 2018? Could this be the reason for changed CH4 fluxes?

**Although we agree there is the possibility of vegetation succession between 2017 and 2018, we don't think the amount of vegetation cover changed significantly between both years. We do however note the small presence of graminoid cover in the MB and SB hollows compared to zero percent cover in the UB hollows. This is referenced in on Pg 7 new lines 13-15 where we indicate an increase in $CH_4$ emissions could be attributed to plant mediated transport.**

Page 4, line 5-7: Why did you start with the gas collections 7 minutes after chamber was placed?

**This method also allows for multiple chambers to be measured at once, therefore increasing our sample size for statistical analyses and has been standard protocol within our research group for many years and has been used in many publications, see:**

- **Strack et al. 2017, Effect of plant functional type on methane dynamics in a restored minerotrophic peatland, Plant Soil, 410: 231-246.**

- **Murray et al. 2017, Methane emissions dynamics from a constructed fen and reference sites in the Athabasca Oil Sands Region, Alberta, Science of the Total Environment, 583: 369-381.**

- **Strack et al. 2018, Impact of winter roads on boreal peatland carbon exchange, Global Change Biology, DOI; 10.1111/gcb/1.3844.**

Page 5, line 7: You have used the soil temperatures measured at 30 cm depth, although measurements were done from 2-30 cm depth. Is there any scientific explanation for that?

**We used 30 cm depth in the analysis as we found it an appropriate depth given the depth of burn.**

Results:
Page 6, line 4-6: Please move this sentence to the Material part. This is partly explaining my question about the conditions in peatlands interior and margin areas.

**This has been moved to Pg 4 new lines 23-25.**

Discussion:
Page 7, line 2: "lend itself to" sound like informal language to me, but as I am not a native speaker, I might be also mistaken.

**We have changed the wording, see Pg 7 new line 26.**

**"The presence of graminoids in the SB hollows post-fire was also found, which could also lead to increasing CH4 emissions in the future, as plant-mediated transport of CH4 is well documented across peatland ecosystems (Bellisario et al., 2016)."**

Page 7, line 19: I would also add the time factor to the sentence. I think "whether" the CH4 emissions will return to similar levels is rather sure. It is much more interesting how long it takes…

**We have re-worded this sentence to make reference to how long it will take for methane emissions to recover to a similar level as the unburned site rather than IF they will recover.**

Figure 2: Graphics of current figure do not able to understand the microform type for most of burn severity classes.

**We have amended the figure;**

---

## Author Comment (AC3) · 4 Jun 2019

**We thank the reviewer for their constructive comments on our manuscript. Here we provide our initial responses to these comments and will provide a modified manuscript after the discussion is closed.**

Davidson and colleagues present a study in which variability in methane emissions have been quantified for different burn severity classes and microtopographic positions in a boreal fen in Canada. The research begins to answer important questions regarding interactions between fire disturbance and methane cycling in the context of boreal carbon cycle feedbacks to climate. The authors find that fire generally reduces methane emissions and, for at least several years following fire, eliminates relationships between water table depth and methane emissions. The paper is based on a relatively small but important data set that is appropriately analyzed. The paper will be suitable for publication after a few relatively minor revisions.

**We thank the reviewer for their positive comments on our study and we outline the detailed reply to their comments below:**

I have several overarching comments followed by more specific ones. This manuscript is rather short, which isn't a bad thing, however I do think there is room to expand and add some additional details, especially in the discussion.

For example, permafrost is invoked as a potentially important driver in the Introduction, but then is not mentioned in the discussion.

**We do not mention permafrost in the discussion as our study site is not underlain by permafrost and we did not want to cause confusion to the reader.**

The main point regarding the wildfire overriding hydrological controls on CH4 emissions comes through. But the secondary point that these effects are understudied and may vary with ecosystem type could be developed more.

I would encourage the authors to archive their data and code in open access repositories, ideally where they would be citable with a doi.

**We can include a Data Availability Statement and state the data that support the findings of this study are available from the corresponding author upon reasonable request.**

It is a little confusing to have CH4 flux numbers reported normally in the text of the results, and then log-transformed in the figures. It would be best to back-transform for the figures if possible.

**As the statistical differences reported in the text are based on analyses using the log-transformed data, we believe it is acceptable to present this data in the figures.**

**However, we now include figures with the untransformed data in the supplementary information for comparison:**

[Figure]

**Figure S3. Methane (CH₄) emissions at each microform type across the peat burn severity gradient for 2017 and 2018. UB is unburned, MB is moderately burned and SB is severely burned.**

[Figure]

**Figure S4. Seasonal mean methane (CH₄) flux at each collar across the peat burn severity gradient plotted against seasonal mean water table (WT) depth. UB is unburned, MB is moderately burned and SB is severely burned.**

P1 L1: Different disciplines may understand peatland methane emissions in different ways. Why not use "Wildfire overrides hydrological controls on boreal peatland methane emissions" or something similarly specific?

**We agree with this comment and have changed the title accordingly.**

P1 L8: Are you referring specifically to a negative carbon cycle feedback to climate, or any negative feedback (e.g. surface energy partitioning)

**We are referring to any negative feedback including impacts on carbon cycle and surface energy partioning for example.**

P1 L19: This sentence seems a little odd since you have not yet mentioned any argument regarding the overriding influence of fire. Maybe reword, or set it up better.

**We have removed this sentence for clarity.**

P2 L13: It would be nice to have the briefest of descriptions of hummocks and hollows, I think I know what they are.

**We included descriptions of hummocks and hollows on Pg 2 new lines 21-27:**

**"Microtopography across peatlands can be impacted through fire, by increasing the prominence of hollows (low lying areas close to the water table; Belyea & Clymo, 1983) on the landscape through altering elevation (Benscoter et al., 2015), and often hollows will have a higher severity of burn compared to other areas across the landscapes (Mayner et al., 2018; Benscoter et al., 2005). Conversely, hummocks (mounded topography, approximately 0.2 m or higher above the water table; Belyea & Clymo, 1983) are generally resistant to fire, namely due to moisture retention differences between the different moss species present at both microform types, as Sphagnum spp. is much more resilient to fire than feather moss (Kettridge et al., 2015)."**

Also what is the mechanism by which fire alters elevation, causing hummocks.

**The fire can remove vegetation and substrate within the hollows much more readily than the fire-resistant Sphagnum hummocks due to the moisture retention differences of the mosses species found at each microform type (Pg 2, new lines 25-27). This vegetation and substrate removal may indeed make the hummocks look more prominent on the landscape, but the fire does not create hummocks.**

Lastly, it would be interesting to know how prevalent hummocks vs. hollows are on the landscape – are there any papers out there with numbers you could site (e.g. hummocks make up XX% and hollows XX% of typical fen's in this region).

**At Poplar Fen, it is estimated that the landscape consists of 47% hummocks and 53% hollows (Gabrielli, 2016, Graduate Thesis, Wilfrid Laurier University) now included on Pg 3, new lines 20-21.**

P2 L17: Can you say specifically which moss species is found at each microform type?

**The dominant hummock forming moss species at this site is *Sphagnum fuscum* which we state on Pg 6 new lines 4-5.**

P2 L31: Is your site underlain by permafrost? I don't think it's reported in the methods. That could be useful to know.

**No, our site is not underlain by permafrost.**

P3 L1: A single site in both Canada and Russia?

**We have clarified in the text that we meant multiple studies by the same author (Köster et al. 2017 and 2018; one in Canada and one in Russia), Pg 3 new lines 4-5.**

P3 L18: Please give a one or two sentence description of the DOB protocol. The reader should only be obligated to look at other refs if they want all of the juicy details.

**We have included a summary of the DOB protocol on Pg 3 new lines 25-29.**

**"The DOB was determined following the protocols used by Lukenbach et al., (2015a), van Beest et al. (In Review). In summary, this method assumes a pre-fire flat surface between multiple reference points across the site, including adventitious roots in the burned sites and unburned reference points. A string is attached between two reference points and ten measurements were taken along the length, from string to burned ground surface, giving an estimate of the depth of the burn."**

P30 L30: Is it possible that these cover variables could have changed of the course of your study? Particularly moss colonization, but also water, which I imagine could change with the weather conditions. Also, could you briefly describe your percent cover approach; since some sites have more than 100% I assume you are looking at over story and ground cover?

**We agree that there is a possibility that vegetation cover changed between 2017 and 2018, however we think the changes would be small and are confident with our vegetation cover values.**

**We did look at ground cover and over story (bryophytes vs. vascular plants) but also would like to highlight the MB and SB sites contained burned areas and bare ground, hence the amount exceeding 100%.**

P4 L12: Please include justification or reasoning for the -5 mg Ch4/m2/d threshold.

**We use the -5 mg $CH_4/m^2/d$ threshold is because we believe it is unlikely for this system to have consumption of methane greater than**

**this. This resulted in only a loss of 2% of the data (6% loss overall after all data checking).**

**Now added to Pg 4 new lines 21-23.**

P6 L25: Do both of these studies indicate both of these things, or is each point from one of the studies?

**We apologise for the confusion. We have clarified in the text that each point is one of the studies, Pg 7 new lines 3-5;**

**"These results contrast with other studies looking at CH4 emissions post-fire at peatland sites, with Danilova et al. (2015) indicating that fire across an ombrotrophic bog could decrease CH4 oxidation due to removal of the methanotrophic community, while and Grau-andrés et al. (2019) note a potential increase CH4 emissions due to increased graminoid cover."**

P7 L1: What do you mean by an addition here?

**We have changed the wording to presence instead of addition, Pg 7 new line 13.**

P7 L4: Please expand the discussion of fire effects on water table depth. A more in depth process level discussion would be nice here, perhaps also with some specifics on the variability within this fen that you allude to.

**We expand on the link between water table, fire and methane production/emissions on Pg 7, new lines 16-31.**

**"Higher emissions at the UB site could result from overall shallower WT at this location compared to the MB and SB sites (Table 1), which were located at the fen margins. Poplar Fen has a highly variable connection to groundwater (Elmes et al., 2018) and the hydrogeologic setting of Poplar Fen likely contributed to the limited effect of the wildfire at this location, but could also result in higher CH₄ emissions than would have occurred naturally at the burned sites prior to the fire. However, the comparison of our results to emissions measured between 2011 and 2014 at another location in Poplar Fen burned during the fire indicate there was no significant difference in CH₄ emissions. Interestingly, we see no relationship with CH₄ emissions and WT depth at the burned sites. This switch in the typical understanding of the relationship between CH₄ emissions and WT further strengthens our argument on the overriding influence of fire. Even under suitable hydrological conditions, there is a lack of CH₄ production, as shown in the incubation study. Removal of vegetation and soil organic matter can lead to drier conditions (Thompson & Waddington, 2013), with a lower water table creating a larger aerobic zone, potentially leading to lower rates of CH₄ production and potentially greater rates of CH₄ consumption. However, fire can also cause a higher water table, which could potentially lead to larger anaerobic zones and potentially higher CH₄ emissions. However, this is dependent on the severity of the burn, where a low severity fire**

**which only removes vegetation and does not impact the microbial community and organic matter content of the soil may still allow for CH$_4$ production. Conversely, a high severity burn which has removed these communities and organic matter may no longer allow for CH$_4$ production, even with suitable hydrological conditions."**

P7 L13: The resistance of S fuscum to what?

**We have clarified on Pg 7 new line 33 that we meant resistance of S. fuscum to fire.**

P7 L14: Chemical changes in the soil substrate? Can you be a little more specific here?

**We have clarified on Pg 7 new line 33 that we meant chemical changes in the soil substrate.**